# The added value of right ventricular function normalized for afterload to improve risk stratification of patients with pulmonary arterial hypertension

**Marco Vicenzi**[1,2,3]*, **Sergio Caravita**[4,5], **Irene Rota**[2], **Rosa Casella**[1], **Gael Deboeck**[6], **Lorenzo Beretta**[7], **Andrea Lombi**[8], **Jean-Luc Vachiery**[3]

**1** Dyspnea Lab, Department of Clinical Sciences and Community Health, University of Milan, Milan, Italy, **2** Cardiovascular Diseases Unit, Fondazione IRCCS Ca' Granda Ospedale Maggiore Policlinico of Milan, Milan, Italy, **3** Department of Cardiology, Cliniques Universitaires de Bruxelles, Hôpital Académique Erasme, Bruxelles, Belgium, **4** Istituto Auxologico Italiano, IRCCS, Ospedale San Luca, Milano, Italy, **5** Department of Management, Information and Production Engineering, University of Bergamo, Dalmine (BG), Italy, **6** Faculty of Motricity Sciences, Department of Physiotherapy, Université Libre de Bruxelles, Bruxelles, Belgium, **7** Scleroderma Unit, Referral Center for Systemic Autoimmune Diseases, Fondazione IRCCS Ca' Granda Ospedale Maggiore Policlinico di Milano, Milan, Italy, **8** Department of Health Science, Pulmonology Unit, University of Milan Bicocca, San Gerardo Hospital, Monza, Italy

* marco.vicenzi@unimi.it

**Data Availability Statement:** All relevant data are available on Zenodo (DOI: 10.5281/zenodo. 5574447).

## Abstract

### Background

Risk stratification is central to the management of pulmonary arterial hypertension (PAH). For this purpose, multiparametric tools have been developed, including the ESC/ERS risk score and its simplified versions derived from large database analysis such as the COMPERA and the French Pulmonary Hypertension Network (FPHN) registries. However, the distinction between high and intermediate-risk profiles may be difficult as the latter lacks granularity. In addition, neither COMPERA or FPHN strategies included imaging-derived markers. We thus aimed at investigating whether surrogate echocardiographic markers of right ventricular (RV) to pulmonary artery (PA) coupling could improve risk stratification in patients at intermediate-risk.

### Material and methods

A single-center retrospective analysis including 102 patients with a diagnosis of PAH was performed. COMPERA and FPHN strategies were applied to stratify clinical risk. The univariate linear regression was used to test the influence of the echo-derived parameters qualifying the right heart (right ventricle basal diameter, right atrial area, and pressure, tricuspid regurgitation velocity, tricuspid annular plane systolic excursion -TAPSE-). Among these, the TAPSE and tricuspid regurgitation velocity ratio (TAPSE/TRV) as well as the TAPSE and systolic pulmonary artery pressure ratio (TAPSE/sPAP) were considered as surrogate of RV-PA coupling.

**Funding:** This work is supported by the "European Respiratory Society Long-Term Research Fellowship (LTRF 94-2012, awarded to MV), the "Belgium Actelion" research grant for the year 2013 (awarded to MV), and the ERS PAH Short-Term Research Training Fellowship (STRTF 2014-5264, awarded to SC) supported by an unrestricted grant by GSK and of the international grant "Cesare Bartorelli" for the year 2014 funded by the Italian Society of Hypertension.

**Competing interests:** The authors have declared that no competing interests exist.

## Results

TAPSE/TRV and TAPSE/sPAP resulted the more powerful markers of prognosis. Once added to COMPERA, TAPSE/TRV or TAPSE/sPAP significantly dichotomized intermediate-risk group in intermediate-to-low-risk (TAPSE/TRV$\geq$3.74 mm·nm/s)$^{-1}$ or TAPSE/sPAP$\geq$0.24 mm/mmHg) and in intermediate-to-high-risk subgroups (TAPSE/TRV<3.74 mm·(m/s)$^{-1}$ or TAPSE/sPAP<0.24 mm/mmHg). In the same way, TAPSE/TRV or TAPSE/sPAP was able to select patients at lower risk among those with 2, 1, and 0 low-risk criteria of both invasive and non-invasive FPHN registries.

## Conclusions

Our results suggest that adopting functional-hemodynamic echo-derived parameters may provide a more accurate risk stratification in patients with PAH. In particular, TAPSE/TRV or TAPSE/sPAP improved risk stratification in patients at intermediate-risk, that otherwise would have remained less characterized.

## Introduction

Pulmonary arterial hypertension (PAH) is a rare and devastating disease [1]. Despite advances in therapy, the prognosis of patients with PAH remains poor, with overall higher mortality rates as compared with other cardiovascular diseases [2,3]. Therefore, multiparametric assessment of patients' risk, both at baseline and during follow-up, plays a pivotal role to set up and implement tailored management strategies on individual basis [4]. Over the past 10 years, several tools have been developed for this purpose. The 2015 ESC/ERS PH guidelines have proposed a semi-quantitative risk stratification, based on several clinical, functional, and hemodynamic parameters. Based on such algorithm, patients are subdivided in three risk categories: low-, intermediate- and high-risk [1]. More recently, simplified risk assessment strategies have been derived from REVEAL, Swedish PAH, COMPERA and French Pulmonary Hypertension Network (FPHN) registries. These newly developed algorithms pragmatically included a limited number of variables, thus overcoming the relative redundancy of the ESC/ERS risk assessment strategy [3,5–7]. In particular, the European COMPERA and FPHN registries have validated a shorter version of the risk score proposed by the ESC/ERS PH guidelines, based on two opposed strategies ("score and average" and "low-risk focused", respectively), both of which proved to accurately predict mortality in patients with PAH [8]. In the COMPERA registry, WHO FC, 6-minute walking distance (6MWD), BNP or NT-proBNP, right atrial pressure (RAP), cardiac index (CI) and mixed venous oxygen saturation were considered. Individual risk was calculated by assigning a score of 1, 2, or 3 to each criterion (1 = low risk, 2 = intermediate risk, and 3 = high risk), and rounding to the mean of the considered variables [6]. The invasive FPHN registry classified at low-risk patients with FC I or II, 6MWD >440 m, RAP <8 mm Hg, and CI $\geq$2.5 L/min/m$^2$. The non-invasive FPHN score replaced the last two hemodynamic parameters with NT-proBNP (<300 ng/l) [7].

Despite the accuracy of these registries in defining prognosis, their precision may be suboptimal especially in the broad category of the intermediate risk profile, where patients may present with "intermediate-low" and "intermediate-high" risk [4].

Echocardiography is a cost-effective, easily available, and non-invasive imaging tool for the detailed assessment of both RV structure and function. Recently, it has been proposed to better

stratify risk in PAH patients once added to REVEAL Lite 2.0 strategy [9]. Interestingly, COM-PERA and FPHN strategies did not include echocardiographic descriptors of right ventricular (RV) function, despite the strong association between RV failure and death in PAH [10]. Tricuspid annular plane systolic excursion (TAPSE) has been suggested to be a reliable echo-derived predictor of event [11–14]. Since RV function in PAH is tightly linked to its afterload, it is appropriate to consider RV and the pulmonary artery (PA) as a synergic unit [15]. In this view, recent studies have tested the clinical relevance of combined echocardiographic parameters surrogate for RV-PA coupling. In particular, the ratio between TAPSE and systolic pulmonary arterial pressure has been validated as a surrogate non-invasive marker for pulmonary-arterial coupling in patients with PAH [16,17]. This is consistent with the pathophysiological dependency of right ventricular function from its afterload [18]. Although it may be useful to predict prognosis, TAPSE/sPAP ratio alone has not been tested to enhance risk stratification in patients falling in the intermediate-risk category. Moreover, we cannot exclude that TAPSE/sPAP performance may be negatively affected by the inaccurate estimation of right atrial pressure (RAP) via vena cava dimension and collapsibility, especially in patients with advanced PAH [19].

In this study, we aimed to assess whether echo-derived right heart parameters can improve risk stratification of patients with PAH. In addition, due to the limitations of RAP estimation via echocardiography, we sought to test the prognostic value of a simplified approach, using TAPSE and tricuspid regurgitation velocity ratio (TAPSE/TRV) instead of TAPSE/sPAP.

## Materials and methods

### Design of the study

The study complied with the Declaration of Helsinki and was approved by the Erasmus Hospital Institutional Review Board (Ref num: P 2012/352). The informed consent was obtained according to the ethics committee's indications.

We retrospectively included consecutive patients with idiopathic PAH (IPAH), heritable PAH, and associated forms of PAH (APAH) who had their first clinical assessment in between June 2000 and June 2015 at the PH Clinic of the Erasme Academic Hospital in Brussels.

Patients were excluded if they had unreadable or unavailable echocardiographic images in the digital archive, or incomplete data for any category of interest: clinical history, six-minute walking test (6MWT), cardiopulmonary exercise test (CPET), right heart catheterization (RHC) at the time of the first complete clinical assessment. Additionally, patients were excluded if they were lost at follow-up. Once the database has been completed, it was anonymized before analysis.

The outcome was defined as a composite of death and lung transplantation (L-Tx).

### Doppler echocardiography

Doppler echocardiography was realized at rest with a standard ultrasound system (Sonos 5500 Ultrasound, from 2000 to 2003; IE33, Philips, Netherlands and Vivid 7 GE Ultrasound, Norway, from 2003 to 2015). A single experienced cardiologist reviewed and analyzed stored images through dedicated software (Xcelera R4.1, Philips Medical Systems), according to recommendations [20].

Right heart function was considered in terms of contractility and remodeling, the former expressed by TAPSE, the latter by the RV dimension (RVEDD, mm), and the right atrial area (RAA, cm$^2$). Doppler sampling of tricuspid regurgitation velocity (TRV, m/s) was used to derive the right ventriculo-atrial systolic pressure gradient through the simplified Bernoulli's equation. Right atrial pressure (RAP) was estimated from the dimension and inspiratory

collapse of the inferior vena cava, according to the current recommendations [21]. Non-invasive sPAP was obtained adding RAP to the right ventriculo-atrial systolic pressure gradient. The efficiency of the cardiopulmonary unit was expressed through the TAPSE/sPAP (mm/mmHg) and the TAPSE/tricuspid regurgitation velocity (TAPSE/TRV, mm·(m/s)$^{-1}$).

### Six-minute walk test

Six-minute walk test was performed in all the patients in a standardized fashion [22]. Patients were instructed to walk back and forth along a 35-m corridor to cover as much ground as possible during 6 minutes. Patients performed two tests on at 2 separate days; the first test was discarded and the second was used for this present study.

### Right heart catheterization

All RHCs were performed by a cardiologist expert in PH, according to standard techniques. The transducer was zeroed at the midthoracic line in a supine patient, halfway between the anterior sternum and the bed surface. All pressure traces were printed at a paper speed of 12.5 mm/sec and read off-line by the same operator. Pulmonary artery pressures (PAP) were measured at end-expiration and averaged over several cardiac cycles (5 to 8). PAWP was measured at mid-A wave. In case of atrial fibrillation, PAWP was measured 130–160 ms after the onset of QRS wave at electrocardiogram and before the v-wave [23]. Cardiac output (CO) was measured by thermodilution in triplicate (using an average of three measurements within 10% of agreement). PVR was calculated as (mean PAP–PAWP)/CO. Pulmonary arterial compliance (Ca) was estimated as the ratio between stroke volume (SV) and pulmonary arterial pulse pressure (PP), and the resistance-compliance product (RC-time) as the product of Ca and modified PVR. During the RHC blood samples were collected to determine mixed oxygen (SvO$_2$) and arterial oxygen saturation (SaO$_2$).

### Cardiopulmonary exercise test (CPET)

A standard, incremental, symptom-limited CPET was performed and interpreted as previously described [24]. Key CPET variables included in the analysis were both oxygen consumption at the peak of exercise (Peak VO$_2$) and the slope of the relationship between minute ventilation and carbon dioxide production (VE/VCO$_2$ slope).

### Risk stratification

Patients were classified according to three risk assessment methods: 1) COMPERA registry strategy, 2) invasive, and 3) non-invasive method of FPHN registry [6,7].

To test the added value of RV function in patients at intermediate-risk according to the COMPERA strategy, we re-assigned each patient to two different sub-groups according to the optimal cut-point of significant echo-derived parameters (see "Statistics" section below). Patients presenting 0, 1, or 2 low-risk criteria of invasive and non-invasive FPHN scores (i.e. non-low-risk patients) were similarly re-classified based on RV function.

### Statistics

All continuous data with normal distribution have been summarised by mean ±standard deviation, those with abnormal distribution by a median (first—third interquartile). Categorical variables have been expressed as frequencies.

A COX univariate linear regression model was used to test the influence on time-to-event of the parameter qualifying the right heart: RVEDD, RAA, TAPSE, TAPSE/sPAP ratio, and TAPSE/TRV.

Parameters significant at the 0.05 threshold were included in Contal and O'Quingley's analysis to stratify the risk of events, calculate the optimal cut-point, and define the corresponding corrected p values [25]. Then, ROC analysis was performed to obtain AUC, sensitivity, and specificity of parameters for optimal cut-point.

The corresponding survival curves were calculated by the Kaplan-Meyer method and p values derived with the log-rank test or Wilcoxon-Breslow-Gehan method.

General characteristics, morphological, functional, and hemodynamic parameters have been included in the sub-group analysis. Intergroup differences were assessed through T-test and Mann-Whitney U test, according to the distribution of variables.

All the analyses except cut-point estimation were performed using IBM SPSS Statistics 24.0.0, Inc., Chicago, IL. Cut-point estimation through Contal and O'Quingley analysis was performed via a custom code written in Python by LB.

## Results

Out of 109 consecutive patients with PAH, 102 met the inclusion criteria. General characteristics of the population are reported in Table 1. IPAH, hereditary and drug/toxin-induced PAH represented the most prevalent aetiology (58%). The great majority of patients were in NYHA FC III-IV (72.5%), the median NT-proBNP value was 1074 pg/ml and the median 6MWT

**Table 1. General characteristics of the patients' population (n 102).**

| General characteristics | |
|---|---|
| Age, y | 54 ±16 |
| Female gender, n (%) | 64 (62.7) |
| BMI, kg/m$^2$ | 23.8 (21.2–28.1) |
| I-II NYHA FC, n (%)<br>III-IV NYHA FC, n (%) | 28 (27.5)<br>74 (72.5) |
| NT-proBNP, pg/ml | 1076.5 (293.5–2447.3) |
| 6MWT distance, m | 415 (302–485.5) |
| PAH aetiology, n (%)<br>• IPAH<br>• Heritable and drug/toxin-induced PAH<br>• APAH<br>• CTD, n (% of APAH)<br>• Cirrhosis, n (% of APAH)<br>• Cardiac shunt, n (% of APAH)<br>• Other, n (% of APAH) | <br>50 (49)<br>9 (8.8)<br>43 (42.2)<br>15 (34.9)<br>10 (23.3)<br>14 (32.6)<br>4 (9.3) |
| PAH-specific treatment, n (%)<br>• Naive<br>• Under treatment<br> • Monotherapy therapy<br> • Combination therapy<br>• ERA<br>• PDE5-i<br>• Prostanoids<br>• CCB<br>• Diuretics | <br>76 (74.5)<br>26 (25.5)<br>21 (80.8)<br>5 (19.2)<br>15 (14.7)<br>4 (3.9)<br>11 (10.8)<br>1 (0.9)<br>45 (44.1) |

PAH: Pulmonary arterial hypertension; BMI: Body mass index; NYHA FC: New York Heart Association Functional Class; IPAH: Idiopathic PAH; APAH: Associated PAH. ERA: Endothelin receptor antagonist; PDE5-i: Phosphodiesterase-5 inhibitors; CCB: Calcium channel blockers.

distance was 415 meters. Seventy-six PAH patients (74.5%) were treatment-naïve, while the other twenty-six subjects (25.5% of the whole population) were already on single (21 pts, 80.8% of treated group) or combined (5 patients, 19.2% of treated group) specific treatment for PAH. Of note, eleven prevalent patients were treated with prostanoids (10 with intravenous and 1 with inhaled prostanoids).

Echocardiographic data are shown in Table 2. On average, our patients presented with right heart remodelling, as reflected by RV dilation (mean RVEDD 47 mm) and RA dilation (mean RAA 25 cm$^2$), with preserved RV systolic function (mean TAPSE 18 mm). Median TRV was 4.2 m/s, which correspond to a systolic right atrial-ventricular gradient of 69 mmHg. Mean TAPSE/TRV and median TAPSE/sPAP ratio were respectively 4.4 mm·(m/s)$^{-1}$ and 0.24 mm/mmHg.

Gas-exchange measures derived through CPET documented a moderate-to-severe reduction of functional capacity and impaired ventilatory efficiency (S1 Table).

Complete hemodynamic data were available for all 102 patients and results are detailed in S1 Table. This demonstrated high PVR (10.8 UW) with reduced CI (2.2 L/min/m$^2$) and low Ca (1.07 mL/mmHg).

The mean follow-up period was 64.6 ±32.6 months: 37 patients died between 2.5 and 82.1 months and 5 underwent lung transplantation (minimum and maximum delay 25.4 and 83.2 months, respectively).

Among the echo-derived right heart indexes, TAPSE, TAPSE/sPAP, and TAPSE/TRV resulted as significant predictors of outcomes in the COX univariate linear regression (Table 3). Through Contal and O'Quingley's analysis, we retested these parameters and calculated the**ir** optimal cut-point: finally, only TAPSE/TRV resulted as significant predictor (T = 3.74 mm·(m/s)$^{-1}$, q = 1.362, p = 0.041).

As shown in Fig 1, risk stratification through the COMPERA strategy was able to differentiate patients at low-risk from those at high-risk for death or L-Tx (respectively, 1-year outcome rate of 0% and 8.5%, log-rank test p = 0.030, Chi$^2$ = 4.697). Patients at intermediate-risk were

**Table 2. Echocardiographic data (n 102).**

| Echocardiography data | |
|---|---|
| LV Ejection fraction, % | 55 ±10 |
| Diastolic pattern, n (%)<br>• Normal<br>• Abnormal Relaxation<br>• Pseudonormal<br>• Atrial fibrillation | <br>29 (28.4)<br>67 (65.7)<br>5 (4.9)<br>1 (1.0) |
| RVEDD, mm | 48.1 ±8.5 |
| RAA, cm$^2$ | 26.4 ±7.7 |
| Estimated RAP, mmHg | 8.4 ±3.7 |
| TRV, m/s | 4.1 ±0.6 |
| RV grad, mmHg | 69 ±20 |
| sPAP, mmHg | 75 (64–90) |
| TAPSE, mm | 17 ±4 |
| TAPSE/sPAP, mm/mmHg | 0.24 (0.18–0.30) |
| TAPSE/TRV, mm·(m/s)$^{-1}$ | 4.4 ±1.3 |

LV: Left ventricle; PAH: Pulmonary arterial hypertension; LHD: Left heart disease; RVEDD: Right end-diastolic diameter; RAA: Right atrial area; RAP: Right atrial pressure; TRV: Tricuspid regurgitation velocity; RV grad: Right ventricular gradient; sPAP: Systolic pulmonary arterial pressure; TAPSE: Tricuspid annular plane systolic excursion.

**Table 3. Univariate regression analysis.**

| | COX univariate linear regression | | |
|---|---|---|---|
| | HR | 95% CI | p |
| RAA, cm$^2$ | 1.032 | 0.998–1.068 | 0.062 |
| RVEDD, mm | 1.030 | 0.996–1.065 | 0.083 |
| TAPSE, mm | 0.920 | 0.853–0.993 | 0.032 |
| TAPSE/sPAP, mm/mmHg | 0.015 | 0.000–0.789 | 0.038 |
| TAPSE/TRV, mm·(m/s)$^{-1}$ | 0.740 | 0.565–0.969 | 0.028 |
| | Contal and O'Quingley's analysis | | |
| | T | q | p |
| TAPSE, mm | 14 | 1.098 | 0.232 |
| TAPSE/sPAP, mm/mmHg | 0.24 | 1.325 | 0.057 |
| TAPSE/TRV, mm·(m/s)$^{-1}$ | 3.74 | 1.362 | 0.041 |

The table shows the univariate and multivariate regression analysis for general characteristic and non-invasive data. The results of univariate analysis for invasive data are also detailed. NYHA FC: New York Heart Association functional class; NT-proBNP: N-terminal pro-brain natriuretic peptide; RVEDD: Right ventricular end-diastolic diameter; RAA: Right atrial area; TAPSE: Tricuspid annulus plane systolic excursion; TAPSE/sPAP: Tricuspid annulus plane systolic excursion and systolic pulmonary arterial pressure ratio; TAPSE/TRV: Tricuspid annulus plane systolic excursion and tricuspid regurgitation maximal velocity ratio.

not significantly separated from those at low-risk (log-rank test p = 0.258, Chi$^2$ = 1.278), while showed a significant reduction of events if compared with those at high-risk (log-rank test p = 0.003, Chi$^2$ = 8.969). The re-stratification of the group at intermediate-risk (n = 39/102)

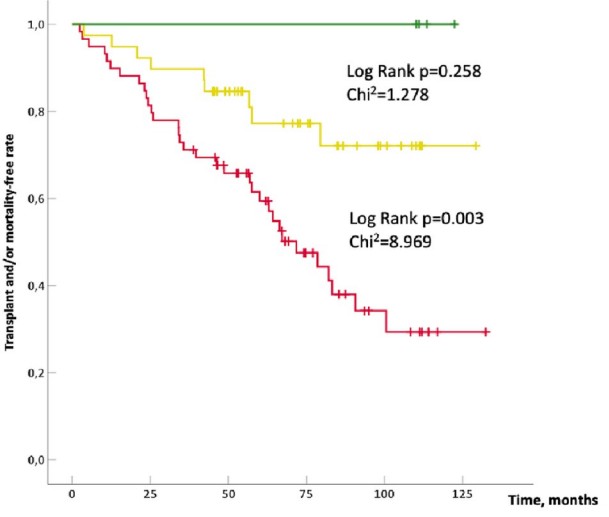

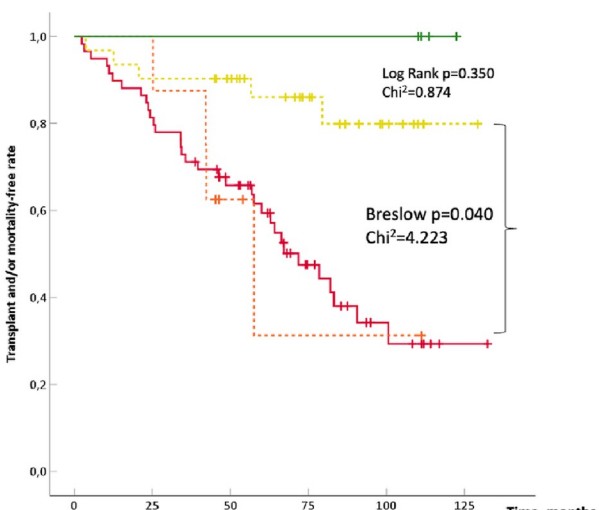

| | Total number of patients at risk and mortality (%) according to group of risk | | | | | |
|---|---|---|---|---|---|---|
| months | 0 | 12 | 24 | 36 | 60 | 120 |
| Low risk, green line | 4 | 4 | 4 | 4 | 4 | 4 |
| | - | 0.0% | 0.0% | 0.0% | 0.0% | 0.0% |
| Intermediate risk, yellow line | 39 | 38 | 36 | 35 | 31 | 30 |
| | - | 2.6% | 7.7% | 10.3% | 20.5% | 23.1% |
| High risk, red line | 59 | 54 | 49 | 42 | 37 | 26 |
| | - | 8.5% | 17.0% | 28.8% | 37.3% | 55.9% |

| | Total number of patients at intermediate risk and mortality (%) according to TAPSE/TRV ratio > or ≤3.74 mmHg*s/m | | | | | |
|---|---|---|---|---|---|---|
| months | 0 | 12 | 24 | 36 | 60 | 120 |
| Intermediate-low risk, yellow line | 31 | 30 | 28 | 28 | 27 | 26 |
| | - | % | 6.7% | 6.7% | 10.0% | 13.3% |
| Intermediate-high risk, orange line | 8 | 8 | 8 | 7 | 4 | 4 |
| | - | 0.0% | 0.0% | 12.5% | 50.0% | 50.0% |

**Fig 1. Kaplan-Meier curves according to PAH risk groups applying the strategy of COMPERA registry.**

based on TAPSE/TRV ($</\geq$3.74 mm·(m/s)$^{-1}$) allowed to distinguish a subgroup of patients at intermediate-to-high risk (n = 8/39) and a subgroup with intermediate-to-low risk (n = 31/39). The former showed a similar event rate to patients at high-risk (Breslow test p = 0.983, Chi$^2$ = 4.223), the latter aggregated patients with not significantly different event rate if compared to those at low-risk (log-rank test p = 0.350, Chi$^2$ = 0.874).

Kaplan-Meyer analysis according to invasive FPHN registry model dichotomized the overall population in two main groups: the first composed by 15 patients with 4 and 3 low-risk criteria (respectively, n = 5/15 and n = 10/15), and the second composed by 87 patients with 2 (n = 29/87), 1 (n = 32/87) or 0 (n = 26/87) low-risk criteria. These two groups presented a significant difference in the rate of adverse events (see Fig 2A, log-rank p = 0.018, Chi$^2$ 5.622). Once TAPSE/TRV was applied to non-low-risk patients (i.e. subjects with 2, 1, or 0 low-risk criteria), two subgroups with different event-free rate curves were differentiated (Fig 2A): subjects with TAPSE/TRV$\geq$3.74 mm·(m/s)$^{-1}$ (n = 55) had a lower outcome rate than those with TAPSE/TRV<3.74 mm·(m/s)$^{-1}$ (1-year event rate of 1.8% vs 15.6%, respectively, log-rank = 0.016 and Chi$^2$ = 5.849). Of note, the subgroup of patients with favourable TAPSE/TRV (orange dashed line in Fig 2A) had a higher but non-significant event rate (log-rank test p = 0.099) compared to the group at lower risk (blue line in Fig 2A). Similar results were obtained with the non-invasive FPHN registry model stratification. Patients with 3 low-risk criteria (green line in Fig 2B) had a non-significantly better prognosis compared to the other three groups (yellow line and blue and red dashed lines). However, once the latter three were grouped together and TAPSE/TRV$\geq$3.74 mm·(m/s)$^{-1}$ was added as a further low-risk marker, two event rate curves were obtained with significant difference (Fig 2B, yellow dashed line vs red dashed line: log-rank test p = 0.003, Chi$^2$ = 9.078).

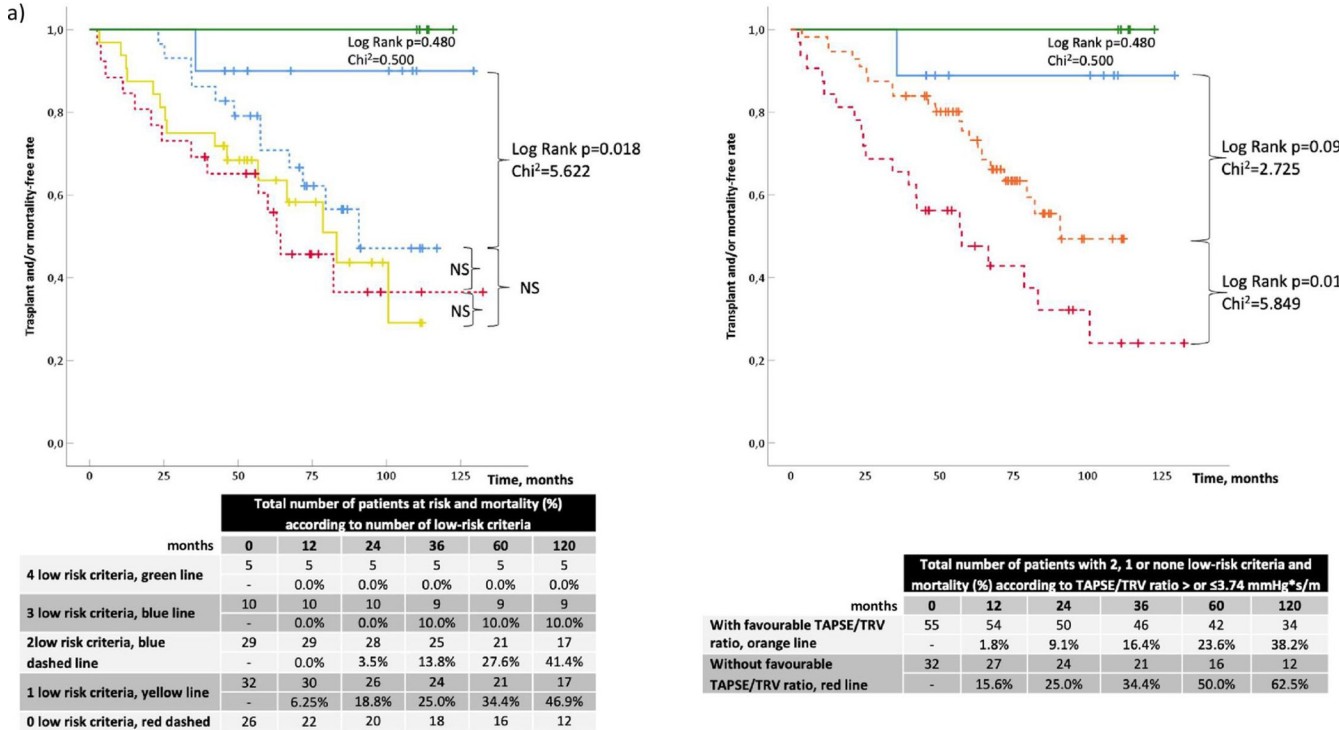

**Fig 2.** Kaplan-Meier curves according to PAH risk groups applying the invasive (a) and non-invasive (b) strategy of FPHN registry.

**Table 4. Characteristics of the population according to TAPSE/TRV ratio values.**

|  | TAPSE/TRV≥3.74 (n 68/102) | TAPSE/TRV<3.74 (n 34/102) | p |
|---|---|---|---|
| Age, yr | 52 ±17 | 57 ±14 | |
| Male gender, n (%) | 28 (40.3) | 10 (31.4) | |
| BMI, kg/m$^2$ | 25.2 (22.1–29.3) | 22.2 (20.4–24.1) | <0.001 |
| NYHA FC, mean | 3 (1–3) | 3 (2–4) | |
| • I-II NYHA FC, n (%) | 15 (22) | 8 (23.5) | |
| • III-IV NYHA FC, n (%) | 53 (78) | 26 (76.5) | |
| NT-proBNP, pg/ml | 732 (172–1809) | 2488 (944–4125) | <0.001 |
| **Echocardiographic variables** | | | |
| • RVEDD, mm | 47.9 ±8.4 | 48.8 ±8.9 | |
| • RAA, cm$^2$ | 26.4 ±8.9 | 26.7 ±5.8 | <0.001 |
| • TAPSE, mm | 19.7 ±3.3 | 13.6 ±3.1 | <0.001 |
| • TRV, m/s | 3.9 ±0.6 | 4.5 ±0.6 | <0.001 |
| • TAPSE/TRV, mm*s/m | 5.2 ±1.2 | 3.0 ±0.5 | |
| **CPET variables** | | | |
| • PeakVO$_2$, ml/kg/min | 12.3 (10.1–15.2) | 11.1 (9–13.3) | <0.01 |
| • VE/VCO$_2$ slope | 52.2 ±18.3 | 62.8 ±24.2 | 0.017 |
| **Invasive variables** | | | |
| mPAP, mmHg | 49 (39–54) | 55 (50–64) | <0.001 |
| RAP, mmHg | 7.4 ±3.6 | 10.9 ±5.5 | <0.001 |
| PAWP, mmHg | 10.0 ±3.3 | 10.1 ±3.3 | 0.015 |
| CI, l/min/m$^2$ | 2.5 ±0.7 | 2.1 ±0.7 | <0.001 |
| PVR, UW | 9.3 ±4.0 | 14.6 ±4.9 | 0.004 |
| SvO$_2$, % | 63.9 ±10.5 | 57.5 ±9.6 | <0.001 |
| SaO$_2$, % | 92.3 ±4.6 | 91.9 ±4.2 | |
| Ca, ml/mmHg | 1.33 (0.97–1.68) | 0.79 (0.68–1.01) | |

According to the cut-point of TAPSE/TRV ratio, the population details are listed in this table and grouped in echocardiography, cardiopulmonary and invasive variables. Statistical significance is reported in the fourth column.

BMI: Body mass index; NYHA FC: New York Heart Association functional class; NT-proBNP: N-terminal pro-brain natriuretic peptide; RVEDD: Right ventricular end-diastolic diameter; RAA: Right atrial area; TAPSE: Tricuspid annulus plane systolic excursion; TRV: Tricuspid regurgitation velocity; TAPSE/TRV: Tricuspid annulus plane systolic excursion and tricuspid regurgitation maximal velocity ratio; PeakVO2: Oxygen consumption at peak of exercise; VE/VCO2 slope: The slope of the relationship between minute ventilation and carbon dioxide production during exercise. mPAP: Mean pulmonary arterial pressure; RAP: Right atrial pressure; PAWP: Pulmonary arterial wedge pressure; CI: Cardiac index; PVR: Pulmonary vascular resistance; SvO2: Mixed venous oxygen saturation; SaO2: Arterial oxygen saturation; Ca: Pulmonary arterial compliance. T-test and Mann-Whitney U test were applied according to the distribution of variables.

In the same way, TAPSE/sPAP was able to identify patients at lower risk in intermediate-risk of COMPERA registry and in groups with 0, 1, or 2 low-risk criteria (see S1 Fig).

When patients were stratified according to the TAPSE/TRV, those with TAPSE/TRV<3.74 mm·(m/s)$^{-1}$ had more impaired functional capacity, higher exercise hyperventilation, as well as a worse hemodynamic profile as compared with patients with TAPSE/TRV≥3.74 mm·(m/s)$^{-1}$, despite similar symptoms burden (Table 4).

The area under the curve at ROC analysis was larger for TAPSE/sPAP (AUC 0.636, p = 0.020) and for TAPSE/TRV (AUC 0.628, p = 0.028) than for TAPSE (AUC 0.588, p = 0.130) (Fig 3). Among the former two parameters, specificity was higher for TAPSE/TRV (0.767) while sensitivity was higher for TAPSE/sPAP (0.714).

## Discussion

The present study showed that a simple combined approach using echo-derived indices can improve risk stratification in patients classified at intermediate-risk through COMPERA

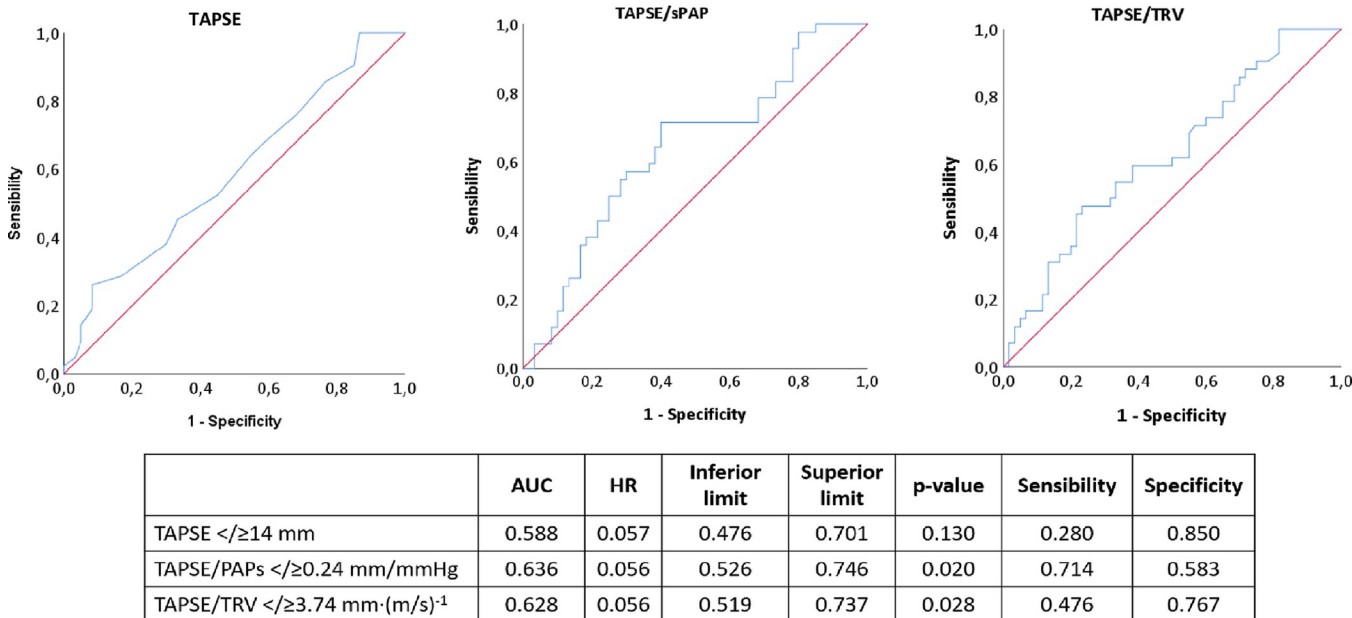

**Fig 3. ROC analysis of TAPSE, TAPSE/sPAP, and TAPSE/TRV according to the optimal cut-point calculated with Contal and O'Quingley analysis.**

|  | AUC | HR | Inferior limit | Superior limit | p-value | Sensibility | Specificity |
|---|---|---|---|---|---|---|---|
| TAPSE </≥14 mm | 0.588 | 0.057 | 0.476 | 0.701 | 0.130 | 0.280 | 0.850 |
| TAPSE/PAPs </≥0.24 mm/mmHg | 0.636 | 0.056 | 0.526 | 0.746 | 0.020 | 0.714 | 0.583 |
| TAPSE/TRV </≥3.74 mm·(m/s)⁻¹ | 0.628 | 0.056 | 0.519 | 0.737 | 0.028 | 0.476 | 0.767 |

strategy and with 0, 1, or 2 low-risk criteria from the FPHN registry. In particular, we could demonstrate that TAPSE/TRV ($\geq$3.74 mm·(m/s)$^{-1}$) and/or TAPSE/sPAP ($\geq$0.24 mm/mmHg) allowed to further stratify intermediate-risk patients, identifying a subgroup of patients at intermediate-low risk, who presented a better prognosis. Indeed, these patients had not-statistically different event free-rate at 1 year of follow-up if compared to those at low-risk according to the ESC/ERS stratification [1]. On the other hand, PAH patients with low values of TAPSE/TRV showed the same events free-rate than reported for subjects at high-risk when stratified according to COMPERA and FPHN registries. Accordingly, our data support the role of the non-invasive assessment of the cardiopulmonary unit as a reliable bedside approach to predict survival and the need for L-Tx. In this context, this study adds to previous evidence supporting the association between TAPSE/sPAP and outcome in PAH [16], albeit suggesting that TAPSE/TRV may be a more specific marker than the former. Indeed, the use of TRV rather than sPAP may obviate several potential sources of error in the echocardiographic estimation of pulmonary hemodynamics. Firstly, TRV is squared and multiplied by factor 4. This result is then summed to an estimation of RAP, by analyzing the diameter and collapsibility of the inferior vena cava (IVC). As shown in several studies, this method provides overall inaccurate and imprecise results [19,26]. In particular, in the setting of chronic heart failure, when RAP is above 7 mmHg, the assessment of IVC collapsibility may provide unreliable estimates of RAP [27]. This might help to explain the better specificity and precision of TAPSE/TRV rather than TAPSE/sPAP in the risk re-stratification in PAH.

Although the relationship between TAPSE and RV function has been shown to be linear and to predict outcome in PAH [1,28], TAPSE represents an oversimplification of RV contractile function and its application is limited when patients present with significant tricuspid regurgitation [29]. Indeed, TAPSE may overestimate RV function during the "compensatory" phase, where RV reserve contractility is recruited at rest in response to afterload increase, before overt RV failure [30,31]. Our data thus seem to confirm that TAPSE, having the highest specificity but lowest sensitivity, is one of the strongest predictive parameters in PAH. This is

consistent with a recent paper by Ghio et al. in which normal TAPSE identifies patients with a good prognosis, whereas reduced values may need additional elements (the degree of tricuspid regurgitation and the diameter of IVC) to better define its prognostic impact [10]. This may further explain why the RV contractility "normalized" by hemodynamic indexes (TAPSE/TRV and TAPSE/sPAP) can provide a more precise evaluation of cardiopulmonary system efficiency and RV-PA coupling.

All survival or event free rate analyses reported in the recently published registries on PAH demonstrated that NYHA functional class was a strong predictor of long-term survival [2,3,6,7]. However, we could not find a significant difference of NYHA FC between the two subgroups when separated according to the TAPSE/TRV, which may suggest that a surrogate marker of RV-PA coupling could add more granularity in outcome prediction. Interestingly, patients with worse TAPSE/TRV presented also with worse cardiorespiratory adaptation to exercise and more impaired hemodynamics, despite a similar symptoms burden (see Table 4). Our results reinforce the clinical role of CPET to assess functional capacity in patients with cardiopulmonary diseases, by benefiting from its pathophysiological comprehensive approach and potential indirect evaluation of right heart functional reserve [32]. This may confirm that non-invasive RV-PA coupling is associated with the global functional and hemodynamic status of patients with PAH [33,34]. This observation is valid also when the population is grouped for TAPSE/sPAP$<$/$\geq$0.24 (see S2 Table).

Finally, even though prognosis is generally established based on hemodynamic characterization, the present study could help to simplify and integrate the clinical assessment. Indeed, less impaired hemodynamics was observed in patients with favourable TAPSE/TRV values. In particular, the invasive variables, RAP, CI, and Ca, were confirmed to be the main predictors of events by univariate analysis (S3 Table in Supporting Information) [35] and resulted significantly worse in patients with unfavourable TAPSE/TRV (Table 4).

## Limitations of the study

Some limitations of the present study should be acknowledged. This is a single-center, retrospective study including a relatively small sample of PAH patients, recruited over a quite long time span. Accordingly, the echocardiographic analysis relied on simple parameters (such as TAPSE) rather than on advanced parameters more representative of RV function (such as RV strain and three-dimensional volumes).

## Conclusions

In conclusion, our data suggest that a combined functional-hemodynamic echo-derived parameter may provide a more accurate risk stratification in patients with PAH. In particular, TAPSE/TRV or TAPSE/sPAP enhanced risk stratification in patients at intermediate-to-high risk, that otherwise would have remained less characterized.

## Supporting information

**S1 Fig.** Kaplan-Meier curves according to risk re-stratification for TAPSE/sPAP for (a) COMPERA registry, (b) invasive FPHN and (c) non-invasive FPHN registry.
(DOCX)

**S1 Table. Results of the baseline assessment through exercise cardiopulmonary testing and right heart catheterization.**
(DOCX)

**S2 Table. Characteristics of the population according to TAPSE/sPAP ratio values.**
(DOCX)

**S3 Table. Univariate regression analysis of hemodynamic data.**
(DOCX)

## Author Contributions

**Conceptualization:** Marco Vicenzi, Sergio Caravita, Irene Rota.

**Data curation:** Marco Vicenzi, Gael Deboeck, Andrea Lombi.

**Formal analysis:** Marco Vicenzi, Irene Rota, Rosa Casella, Lorenzo Beretta.

**Investigation:** Marco Vicenzi, Gael Deboeck.

**Methodology:** Marco Vicenzi, Rosa Casella.

**Software:** Gael Deboeck, Lorenzo Beretta.

**Supervision:** Jean-Luc Vachiery.

**Validation:** Irene Rota, Rosa Casella, Gael Deboeck, Lorenzo Beretta, Andrea Lombi, Jean-Luc Vachiery.

**Visualization:** Irene Rota.

**Writing – original draft:** Marco Vicenzi, Lorenzo Beretta.

**Writing – review & editing:** Marco Vicenzi, Sergio Caravita, Jean-Luc Vachiery.

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
