## [Decision Letter · Decision Letter 0]

20 Aug 2021

PONE-D-21-22876

The added value of right ventricular function normalized for afterload to improve risk stratification of patients with pulmonary arterial hypertension

PLOS ONE

Dear Dr. Vicenzi,

Thank you for submitting your manuscript to PLOS ONE. After careful consideration, we feel that it has merit but does not fully meet PLOS ONE’s publication criteria as it currently stands. Therefore, we invite you to submit a revised version of the manuscript that addresses the points raised during the review process.

Both reviewers had areas where they thought clarification was needed, and potentially inclusion of additional data which you're likely to already have in hand.

We look forward to receiving your revised manuscript.

Kind regards,

James West, PhD

Academic Editor

PLOS ONE

Journal Requirements:

2. Please provide additional details regarding participant consent. In the Methods section, please ensure that you have specified (1) whether consent was informed and (2) what type you obtained (for instance, written or verbal). If your study included minors, state whether you obtained consent from parents or guardians. If the need for consent was waived by the ethics committee, please include this information.

3. In the ethics statement in the manuscript and in the online submission form, please provide additional information about the patient records used in your retrospective study, including: a) whether all data were fully anonymized before you accessed them; b) the date range (month and year) during which patients' medical records were accessed; c) the date range (month and year) during which patients whose medical records were selected for this study sought treatment. If the ethics committee waived the need for informed consent, or patients provided informed written consent to have data from their medical records used in research, please include this information.

Reviewers' comments:

Reviewer's Responses to Questions

**Comments to the Author**

1. Is the manuscript technically sound, and do the data support the conclusions?

Reviewer #1: Yes

Reviewer #2: Yes

2. Has the statistical analysis been performed appropriately and rigorously? 

Reviewer #1: Yes

Reviewer #2: Yes

3. Have the authors made all data underlying the findings in their manuscript fully available?

Reviewer #1: Yes

Reviewer #2: Yes

4. Is the manuscript presented in an intelligible fashion and written in standard English?

Reviewer #1: Yes

Reviewer #2: Yes

5. Review Comments to the Author

Reviewer #1: Title : The added value of right ventricular function normalized for afterload to improve risk stratification of patients with pulmonary arterial hypertension

Authors : Vicenzi M et al.

Comments

In the introduction and in abstract, the authors have repeatedly stated that the current risk stratification scores do not include echo cardiographic parameters. I don’t think this statement is totally correct. I agree that the echo/ imaging parameters have not been included as much we want them to be, but REVEAL has pericardial effusion and ESC/ERS has RA area in them. Thus, I would suggest changing the introduction to state something like only limited imaging parameters have been included which do not reflect the comprehensive view of the RV function.

Methods: The study includes patients from over 15 years. There is no mention of how these patients were included, selected or all patients with IPAH or HPAH. Consecutive patients with regular follow up? How did they account for patients who were lost to follow up or whose outcome data was not available?

“All non-invasive tests were performed within 14 days, without changes in treatments or patients’ conditions” this statement need clarity, 14 days of what? 14 days of stable therapy? That’s generally a very short period to consider stability on therapy

Also methods section, describes the methods of each testing ie echo, RHC, walk test etc but it doesn’t describe study design. For example, how these patients were selected? If a patient had multiple serial echo’s and walk test or CPET studies, how did the authors decide which echo to review or which walk test to include or CPET. Or they included all patients with all of their testings?

“To control for any residual learning effect, the second of two tests performed on at least 2 separate days was used for the present study.”

Above statement need clarification. Since it’s a retrospective study, these tests are standard of care, so how did they have walk tests done at two different days? My understanding is, as a part of routine follow ups patients generally perform one walk test.

Results: RHC and CPET data are not directly relevant to the central focus of the study. So I would suggest to put in as supplementary data

Results are not flowing well. It is very confusing in the way its currently written. I understand the message but the presentation is very confusing.

Again, there is no mention that these echo and all other parameters are baseline? Ie diagnostic or the follow ups

How did the authors come up with the cut of 3.74 for TAPSE/TRV ratio?

In the French analysis, without echo, the authors are saying 15 with 4/3 low risk criteria vs 2/1/0 low risk had no difference in outcomes ? explain this.

AUC of the TAPSE/TRV is very low—so without the risk score if someone uses it as stand alone parameter, it doesn’t have much predictive value.

Is it possible for the authors to combine the TAPSE/TRV ratio with the average compera score and compare it with COMPERA score alone to predict outcome

Discussion is nicely written.

Reviewer #2: The article is very good and I am sure that many people will read and use it.

Abstract: Usually, the audience should find out your most important findings by reading the abstract, while you have generalized in the this part and have not mentioned any number or cut point. By inserting this part, the summary of the article will be more complete and better. In addition, your abstract is not structured.

Introduction: If you find it appropriate, explain a little more about the study of COMPERA and the French PNH and its classifications, so that the audience does not have to refer to other data articles to understand the content from the beginning.

Discussion: In the section on discussing the characteristics of people according to the TAPSE / sPAP ratio, although what you are looking for is understandable, but like the data in Table 4, you did not provide a separate table for this ratio and Cutpoint 0.24, and did not explain. which it is appropriate to give a similar tabular data or give an explanation in this regard.

6. PLOS authors have the option to publish the peer review history of their article (what does this mean?). If published, this will include your full peer review and any attached files.

Reviewer #1: **Yes: **Sandeep Sahay

Reviewer #2: No

---

## [Author Response · Author response to Decision Letter 0]

6 Nov 2021

We would thank the reviewers for their thoughtful comments that helped us to revise and hopefully improve our work.

Please find below a point-by-point answers and the respective changes in the manuscript.

Reviewer #1: 

- In the introduction and in abstract, the authors have repeatedly stated that the current risk stratification scores do not include echo cardiographic parameters. I don’t think this statement is totally correct. I agree that the echo/ imaging parameters have not been included as much we want them to be, but REVEAL has pericardial effusion and ESC/ERS has RA area in them. Thus, I would suggest changing the introduction to state something like only limited imaging parameters have been included which do not reflect the comprehensive view of the RV function.

A: We thank the reviewer for this comment. We agree that echocardiography is took in account in REVEAL and in ESC/ERS risk stratification. However, echocardiography is not included in the simplified risk algorithms proposed by COMPERA or the French network. This has been clarified in the text.

- Methods: The study includes patients from over 15 years. There is no mention of how these patients were included, selected or all patients with IPAH or HPAH. Consecutive patients with regular follow up? How did they account for patients who were lost to follow up or whose outcome data was not available?

A: Thank you for this important comment. Our center has used echocardiography and biomarkers since the beginning. We have now detailed the selection of patients.

- “All non-invasive tests were performed within 14 days, without changes in treatments or patients’ conditions” this statement need clarity, 14 days of what? 14 days of stable therapy? That’s generally a very short period to consider stability on therapy

A: This point is now simplified and clarified. All non-invasive tests were executed up to 14 days after changes of medical therapy. We therefore believe this is unlikely to affect our results.

- Also methods section, describes the methods of each testing ie echo, RHC, walk test etc but it doesn’t describe study design. For example, how these patients were selected? If a patient had multiple serial echo’s and walk test or CPET studies, how did the authors decide which echo to review or which walk test to include or CPET. Or they included all patients with all of their testings?

A: Thank you for this comment. The text is now updated, and the patients’ selection is detailed.

- “To control for any residual learning effect, the second of two tests performed on at least 2 separate days was used for the present study.”

Above statement need clarification. Since it’s a retrospective study, these tests are standard of care, so how did they have walk tests done at two different days? My understanding is, as a part of routine follow ups patients generally perform one walk test.

A: We thank the reviewer for this important comment. As stated in a previous response, our center has adopted the standards of clinical trials since the late 90’S. Therefore, all patients underwent a first learning/training test and a second one that was used for the present study. This approach is known to limiting the risk of possible bias. We have clarified the text.

- Results: RHC and CPET data are not directly relevant to the central focus of the study. So I would suggest to put in as supplementary data

A: Thanks for this suggestion. We have built S1 Table.

- Results are not flowing well. It is very confusing in the way its currently written. I understand the message but the presentation is very confusing.

Again, there is no mention that these echo and all other parameters are baseline? Ie diagnostic or the follow ups

How did the authors come up with the cut of 3.74 for TAPSE/TRV ratio?

A: We thank the reviewer for this comment. Optimal cut-point was calculated via Contal and O’Quingley’s analysis. We have now specified in the text.

- In the French analysis, without echo, the authors are saying 15 with 4/3 low risk criteria vs 2/1/0 low risk had no difference in outcomes ? explain this.

A: Thanks for this comment. The FPHN strategy was able to separate patients with 4/3 low criteria from those with 2/1/0 low criteria. As now reported in the text we detailed the results and we specified that no significant difference was found among the two groups with 4 or 3 low criteria and among the three groups with 2, 1 or 0 criteria.

- AUC of the TAPSE/TRV is very low—so without the risk score if someone uses it as stand alone parameter, it doesn’t have much predictive value.

A: Thank you for this important comment. We performed ROC analysis in order to compare TAPSE, TAPSE/TRV and TAPSE/sPAP as marker of contractility, the former, and surrogate of RV-PA coupling, the last two. We agree that AUCs were absolutely low for TAPSE/TRV and TAPSE/sPAP, however relatively to TAPSE they were bigger and obtained a significant p value. In our analysis TAPSE/TRV or TAPSE/sPAP have not been considered as prognostic marker standing alone, but added as “second-step analysis” to COMPERA and FPHN strategies. On the other hand, we did not suggest to use TAPSE/TRV or TAPSE/sPAP alone, and moreover they may be included in a multiparameter approach.

- Is it possible for the authors to combine the TAPSE/TRV ratio with the average compera score and compare it with COMPERA score alone to predict outcome.

A: We thank the reviewer for this suggestion. We performed the analysis as here reported. However, TAPSE/TRV was not able to further separate patient at high-risk (high-high vs high-low, blue line vs green line. Chi2 1.628, log rank 0.202). We did not apply the TAPSE/TRV stratification to group at low-risk because of the absence of events (n=4, survival 100%).

Discussion is nicely written.

We would like to thank the reviewer for her/his appreciation of our manuscript. 

Reviewer #2: 

The article is very good and I am sure that many people will read and use it.

- Abstract: Usually, the audience should find out your most important findings by reading the abstract, while you have generalized in the this part and have not mentioned any number or cut point. By inserting this part, the summary of the article will be more complete and better. In addition, your abstract is not structured.

A: We would like to thank the reviewer for her/his kind words. We have written abstract and structured as requested.

- Introduction: If you find it appropriate, explain a little more about the study of COMPERA and the French PNH and its classifications, so that the audience does not have to refer to other data articles to understand the content from the beginning.

A: The text is now updated: COMPERA and FPHN registries are now better described.

- Discussion: In the section on discussing the characteristics of people according to the TAPSE / sPAP ratio, although what you are looking for is understandable, but like the data in Table 4, you did not provide a separate table for this ratio and Cutpoint 0.24, and did not explain. which it is appropriate to give a similar tabular data or give an explanation in this regard.

A: Thank you for this comment. We built the Table 4 and we discussed the differences among the two subgroups in order to point out that functional capacity assessed via exercise cardiopulmonary testing and not NYHA FC is associated to RV-PA coupling and provides a more sensible and comprehensive approach. 

We added S1 Table built according to the TAPSE/sPAP ratio as suggested.

We would like to thank the reviewer for his appreciation of our manuscript.

---

## [Decision Letter · Decision Letter 1]

27 Dec 2021

PONE-D-21-22876R1The added value of right ventricular function normalized for afterload to improve risk stratification of patients with pulmonary arterial hypertensionPLOS ONE

Dear Dr. Vicenzi,

Thank you for submitting your manuscript to PLOS ONE. After careful consideration, we feel that it has merit but does not fully meet PLOS ONE’s publication criteria as it currently stands. Therefore, we invite you to submit a revised version of the manuscript that addresses the points raised during the review process. There are slight additional details requested.

We look forward to receiving your revised manuscript.

Kind regards,

James West, PhD

Academic Editor

PLOS ONE

Reviewers' comments:

Reviewer's Responses to Questions

**Comments to the Author**

1. If the authors have adequately addressed your comments raised in a previous round of review and you feel that this manuscript is now acceptable for publication, you may indicate that here to bypass the “Comments to the Author” section, enter your conflict of interest statement in the “Confidential to Editor” section, and submit your "Accept" recommendation.

Reviewer #1: (No Response)

Reviewer #2: All comments have been addressed

2. Is the manuscript technically sound, and do the data support the conclusions?

Reviewer #1: Yes

Reviewer #2: Yes

3. Has the statistical analysis been performed appropriately and rigorously? 

Reviewer #1: Yes

Reviewer #2: Yes

4. Have the authors made all data underlying the findings in their manuscript fully available?

Reviewer #1: Yes

Reviewer #2: Yes

5. Is the manuscript presented in an intelligible fashion and written in standard English?

Reviewer #1: No

Reviewer #2: Yes

6. Review Comments to the Author

Reviewer #1: Thank you addressing my comments.

I think your team has answered most of the concerns but your response to reviewer file doesn't reflect that well. I would like you to send responses, citing page number and the line number where you have provided the response. it appears to be missing from response to reviewer file. additionally, how did you define events in this analysis? I think I missed it in earlier version to ask this. For example, see below. Through out the manuscript you are referring to events, what do you mean by these events, are they mean death/transplant or they mean clinical worsening as defined as ....? or PAH related hospitalization or its composite of all of these. its important to define these well

Patients at intermediate-risk were not significantly separated from those at low-risk (log-rank test p=0.258, Chi2=1.278), while showed a significant reduction of events if compared with those at high-risk (log-rank test p=0.003, Chi2=8.969).

In the above statement, what are the events? How did they define it? I don’t see any mention in the methods section.

Additionally, is it possible for the authors to combine the risk score from COMPERA/French scoring to combine with the echo parameters and come up with the ROC/AUC as recently published by Sahay S et al E-REVEALlite 2.0 in predicting disease progression in PAH. published in pulmonary circulation. https://onlinelibrary.wiley.com/doi/abs/10.1002/pul2.12026

Lastly, I think it will be nice to get the finalized paper reviewed by someone who is non-physician English speaker to read through for any grammatical errors and to correct syntax of many statements.

Reviewer #2: Thanks to the respected authors, I think all the points have been accepted and applied. The article is well organized and now has a complete and understandable form. Also, the right conclusions are obtained and well supported by data are and suitable for publication.

7. PLOS authors have the option to publish the peer review history of their article (what does this mean?). If published, this will include your full peer review and any attached files.

Reviewer #1: No

Reviewer #2: **Yes: **Hamidreza Javadi. Associated professor of cardiology. Cardiology department. Faculty of medicine. Qazvin university of medical sciences. Qazvin. IRAN.

---

## [Author Response · Author response to Decision Letter 1]

7 Feb 2022

We would thank the reviewers for their constructive comments. We believe it substantially helped us improving our work, and we would like to apologise for delivering a revision that did not reflect the requested changes. 

Please find below the point by point responses we wish to provide to the reviewer in R1, with the page/line reference for each answer

Reviewer #1: Thank you addressing my comments.

I think your team has answered most of the concerns but your response to reviewer file doesn’t reflect that well. I would like you to send responses, citing page number and the line number where you have provided the response. It appears to be missing from response to reviewer file.

-- We thank the reviewer for his comment. We are sorry for delivering a revised manuscript that did not reflect the requested changes. Please find below the point-by-point responses we wish to provide to the reviewer in R1, with the page/line reference for each answer.

Additionally, how did you define events in this analysis? I think I missed it in earlier version to ask this. For example, see below. Through out the manuscript you are referring to events, what do you mean by these events, are they mean death/transplant or they mean clinical worsening as defined as ....? or PAH related hospitalization or its composite of all of these. Its important to define these well. Patients at intermediate-risk were not significantly separated from those at low-risk (log-rank test p=0.258, Chi2=1.278), while showed a significant reduction of events if compared with those at high-risk (log-rank test p=0.003, Chi2=8.969). In the above statement, what are the events? How did they define it? I don’t see any mention in the methods section.

-- Thank you for stressing this important point. As reported in the text (see section “Materials and Method – design of the study), we considered as event exclusively death and lung transplantation. The patients that were lost at follow-up were not included in analysis (page 6 line 12-14).

Additionally, is it possible for the authors to combine the risk score from COMPERA/French scoring to combine with the echo parameters and come up with the ROC/AUC as recently published by Sahay S et al E-REVEALlite 2.0 in predicting disease progression in PAH. published in pulmonary circulation. 

-- We thank the reviewer for this suggestion and we do acknowledge the very elegant work of Sahay et al, now added the reference. We however feel that it may be inappropriate to multiply the risk assessment tools as they appear to provide consistent answers. We would therefore prefer keeping our message as simple as possible, if this is agreeable to the reviewer

Lastly, I think it will be nice to get the finalized paper reviewed by someone who is non-physician English speaker to read through for any grammatical errors and to correct syntax of many statements.

-- Thank you for your comment. The manuscript was proof-edited by a native English speaker.

Reviewer #2: Thanks to the respected authors, I think all the points have been accepted and applied. The article is well organized and now has a complete and understandable form. Also, the right conclusions are obtained and well supported by data are and suitable for publication.

-- We thank the reviewer for his very positive comment.

4th October 2021 – Responses to Reviewer #1: 

- In the introduction and in abstract, the authors have repeatedly stated that the current risk stratification scores do not include echo cardiographic parameters. I don’t think this statement is totally correct. I agree that the echo/ imaging parameters have not been included as much we want them to be, but REVEAL has pericardial effusion and ESC/ERS has RA area in them. Thus, I would suggest changing the introduction to state something like only limited imaging parameters have been included which do not reflect the comprehensive view of the RV function.

We thank the reviewer for this comment. We agree that echocardiography is took in account in REVEAL and in ESC/ERS risk stratification. However, echocardiography is not included in the simplified risk algorithms proposed by COMPERA or the French network. This has been clarified in the text (see page 4, line 25 in final R2 version)

- Methods: The study includes patients from over 15 years. There is no mention of how these patients were included, selected or all patients with IPAH or HPAH. Consecutive patients with regular follow up? How did they account for patients who were lost to follow up or whose outcome data was not available?

Thank you for this important comment. Our center has used echocardiography and biomarkers since the beginning. We have now detailed the selection of patients (see page 6, from line 6 to line 12 in final R2 version)

- “All non-invasive tests were performed within 14 days, without changes in treatments or patients’ conditions” this statement need clarity, 14 days of what? 14 days of stable therapy? That’s generally a very short period to consider stability on therapy

This point is now simplified and clarified. All non-invasive tests were executed up to 14 days after changes of medical therapy. We therefore believe this is unlikely to affect our results.

- Also methods section, describes the methods of each testing ie echo, RHC, walk test etc but it doesn’t describe study design. For example, how these patients were selected? If a patient had multiple serial echo’s and walk test or CPET studies, how did the authors decide which echo to review or which walk test to include or CPET. Or they included all patients with all of their testings?

Thank you for this comment. The text is now updated, and the patients’ selection is detailed (see page 6, lines 11 and 12)

- “To control for any residual learning effect, the second of two tests performed on at least 2 separate days was used for the present study.”

Above statement need clarification. Since it’s a retrospective study, these tests are standard of care, so how did they have walk tests done at two different days? My understanding is, as a part of routine follow ups patients generally perform one walk test.

We thank the reviewer for this important comment. As stated in a previous response, our center has adopted the standards of clinical trials since the late 90’S. Therefore, all patients underwent a first learning/training test and a second one that was used for the present study. This approach is known to limiting the risk of possible bias. We have clarified the text (see page 7, lines 5 and 6)

- Results: RHC and CPET data are not directly relevant to the central focus of the study. So I would suggest to put in as supplementary data

Thanks for this suggestion. We have built S1 Table.

- Results are not flowing well. It is very confusing in the way its currently written. I understand the message but the presentation is very confusing.

Again, there is no mention that these echo and all other parameters are baseline? Ie diagnostic or the follow ups

How did the authors come up with the cut of 3.74 for TAPSE/TRV ratio?

We thank the reviewer for this comment. Optimal cut-point was calculated via Contal and O’Quingley’s analysis (see page 8 lines 16 and 17 in final R2 version). We have now specified in the text.

- In the French analysis, without echo, the authors are saying 15 with 4/3 low risk criteria vs 2/1/0 low risk had no difference in outcomes ? explain this.

Thanks for this comment. The FPHN strategy was able to separate patients with 4/3 low criteria from those with 2/1/0 low criteria. As now reported in the text we detailed the results and we specified that no significant difference was found among the two groups with 4 or 3 low criteria and among the three groups with 2, 1 or 0 criteria (see page 13, from line 3 to 7)

- AUC of the TAPSE/TRV is very low—so without the risk score if someone uses it as stand alone parameter, it doesn’t have much predictive value.

Thank you for this important comment. We performed ROC analysis in order to compare TAPSE, TAPSE/TRV and TAPSE/sPAP as marker of contractility, the former, and surrogate of RV-PA coupling, the last two. We agree that AUCs were absolutely low for TAPSE/TRV and TAPSE/sPAP, however relatively to TAPSE they were bigger and obtained a significant p value. In our analysis TAPSE/TRV or TAPSE/sPAP have not been considered as prognostic marker standing alone, but added as “second-step analysis” to COMPERA and FPHN strategies. On the other hand, we did not suggest to use TAPSE/TRV or TAPSE/sPAP alone, and moreover they may be included in a multiparameter approach.

- Is it possible for the authors to combine the TAPSE/TRV ratio with the average compera score and compare it with COMPERA score alone to predict outcome.

We thank the reviewer for this suggestion. We performed the analysis as here reported. However, TAPSE/TRV was not able to further separate patient at high-risk (high-high vs high-low, blue line vs green line. Chi2 1.628, log rank 0.202). We did not apply the TAPSE/TRV stratification to group at low-risk because of the absence of events (n=4, survival 100%).

Discussion is nicely written.

We would like to thank the reviewer for her/his appreciation of our manuscript. 

4th October 2021 – Response to Reviewer #2: 

The article is very good and I am sure that many people will read and use it.

- Abstract: Usually, the audience should find out your most important findings by reading the abstract, while you have generalized in the this part and have not mentioned any number or cut point. By inserting this part, the summary of the article will be more complete and better. In addition, your abstract is not structured.

We would like to thank the reviewer for her/his kind words. We have written abstract and structured as requested.

- Introduction: If you find it appropriate, explain a little more about the study of COMPERA and the French PNH and its classifications, so that the audience does not have to refer to other data articles to understand the content from the beginning.

The text is now updated: COMPERA and FPHN registries are now better described.

- Discussion: In the section on discussing the characteristics of people according to the TAPSE / sPAP ratio, although what you are looking for is understandable, but like the data in Table 4, you did not provide a separate table for this ratio and Cutpoint 0.24, and did not explain. which it is appropriate to give a similar tabular data or give an explanation in this regard.

Thank you for this comment. We built the Table 4 and we discussed the differences among the two subgroups in order to point out that functional capacity assessed via exercise cardiopulmonary testing and not NYHA FC is associated to RV-PA coupling and provides a more sensible and comprehensive approach. 

We added S1 Table built according to the TAPSE/sPAP ratio as suggested.

---

## [Decision Letter · Decision Letter 2]

23 Feb 2022

The added value of right ventricular function normalized for afterload to improve risk stratification of patients with pulmonary arterial hypertension

PONE-D-21-22876R2

Dear Dr. Vicenzi,

We’re pleased to inform you that your manuscript has been judged scientifically suitable for publication and will be formally accepted for publication once it meets all outstanding technical requirements. This final version includes all the changes suggested by the reviewers; the text has improved in the various points suggested and enriched the subject matter. 

Kind regards,

Lucio Careddu, Ph.D.

Academic Editor

PLOS ONE

Additional Editor Comments (optional):

Reviewers' comments:

Reviewer's Responses to Questions

**Comments to the Author**

1. If the authors have adequately addressed your comments raised in a previous round of review and you feel that this manuscript is now acceptable for publication, you may indicate that here to bypass the “Comments to the Author” section, enter your conflict of interest statement in the “Confidential to Editor” section, and submit your "Accept" recommendation.

Reviewer #1: All comments have been addressed

Reviewer #2: All comments have been addressed

2. Is the manuscript technically sound, and do the data support the conclusions?

Reviewer #1: Yes

Reviewer #2: Yes

3. Has the statistical analysis been performed appropriately and rigorously? 

Reviewer #1: Yes

Reviewer #2: Yes

4. Have the authors made all data underlying the findings in their manuscript fully available?

Reviewer #1: Yes

Reviewer #2: Yes

5. Is the manuscript presented in an intelligible fashion and written in standard English?

Reviewer #1: Yes

Reviewer #2: Yes

6. Review Comments to the Author

Reviewer #1: Excellent job by the authors in addressing comments! Congratulations. I do not have any further comments

Reviewer #2: Thank you for your good article. The applied changes have made your article better, smoother and more understandable. I suggest you complete the title of Table 3 as well. Univariate and multivariate regression analysis of echocardiographic parameters.

7. PLOS authors have the option to publish the peer review history of their article (what does this mean?). If published, this will include your full peer review and any attached files.

Reviewer #1: No

Reviewer #2: **Yes: **Hamidreza Javadi.MD. Cardiologist. Qazvin University of medical sciences. Qazvin. Iran.

---

## [Editor Report · Acceptance letter]

25 Apr 2022

PONE-D-21-22876R2 

The added value of right ventricular function normalized for afterload to improve risk stratification of patients with pulmonary arterial hypertension 

Dear Dr. Vicenzi:

I'm pleased to inform you that your manuscript has been deemed suitable for publication in PLOS ONE. Congratulations! Your manuscript is now with our production department. 

Kind regards, 

on behalf of

Dr. Lucio Careddu 

Academic Editor

PLOS ONE